# Helmet-Wearing Tracking Detection Based on StrongSORT

**DOI:** 10.3390/s23031682

**Published:** 2023-02-03

**Authors:** Fufang Li, Yan Chen, Ming Hu, Manlin Luo, Guobin Wang

**Affiliations:** School of Computer Science and Cyber Engineering, Guangzhou University, Guangzhou 510006, China

**Keywords:** YOLOv5, focal-EIOU, object detection, StrongSORT, helmet wear tracking

## Abstract

Object detection based on deep learning is one of the most important and fundamental tasks of computer vision. High-performance detection algorithms have been widely used in many practical fields. For the management of workers wearing helmets in construction scenarios, this paper proposes a framework model based on the YOLOv5 detection algorithm, combined with multi-object tracking algorithms, to monitor and track whether workers wear safety helmets in real-time video. The improved StrongSORT tracking algorithm of DeepSORT is selected to reduce the loss of the tracked object caused by the occlusion, trajectory blur, and motion scale of the object. The safety helmet dataset is trained with YOLOv5s, and the best result of training is used as the weight model in the StrongSORT tracking algorithm. The experimental results show that the mAP@0.5 of all classes in the YOLOv5s model can reach 95.1% in the validation dataset, mAP@0.5:0.95 is 62.1%, and the precision of wearing helmet is 95.7%. After the box regression loss function was changed from CIOU to Focal-EIOU, the mAP@0.5 increased to 95.4%, mAP@0.5:0.95 increased to 62.9%, and the precision of wearing helmet increased to 96.5%, which were increased by 0.3%, 0.8% and 0.8%, respectively. StrongSORT can update object trajectories in video frames at a speed of 0.05 s per frame. Based on the improved YOLOv5s combined with the StrongSORT tracking algorithm, the helmet-wearing tracking detection can achieve better performance.

## 1. Introduction and Related Work

The safety of the construction site has always been a hot issue. Wearing a helmet can greatly reduce a heavy blow to the head during the construction process, and is also a basic guarantee for the personal safety of workers. The accidents caused by the lack of safety helmets during construction are still on the rise. Therefore, real-time monitoring of construction sites and management to ensure workers wear safety helmets are effective ways to reduce the occurrence of safety accidents.

The task of real-time monitoring whether workers are wearing a helmet consists of object detection and target tracking.

Object detection has always been a hot research direction of computer vision. In short, its main task is to classify and locate algorithms in specific scenes, and many excellent detection algorithms have also been combined in applications in various fields. Traditional object detection [1,2] is a feature extractor using manual identification. However, these traditional model algorithms are relatively slow in detection speed and poor in detection precision compared with the current algorithms based on deep learning. The generalization effect of the model on the test dataset is not very good, and the performance applied to the actual project cannot meet the specified standards. As research on CNNs (convolutional neural networks) [3] continues to advance, object detection can be divided into two methods: two-stage and one-stage.

Two-stage is an algorithm based on region proposals; first, extracting region proposals, and then performing a classification regression task on the region proposals. Girshick, R. et al. [4] proposed R-CNN (Region CNN) for the first time, and used selective search [5] to extract region proposals. The backbone used the AlexNet [6] network model as the detector. The proposed R-CNN greatly promoted the research in the field of object detection. Later, Girshick, R. and Ren, S. et al. [7,8] proposed Fast R-CNN and Faster R-CNN, respectively, which greatly improved the detection precision. In [8], it is proposed to use the RPN (Region Proposal Network) to generate region proposals instead of selective search. Meanwhile, the idea of generating region proposals based on anchor box is proposed.

The one-stage detection algorithm uses end-to-end training to directly classify and locate objects after feature extraction of samples, such as the YOLO series [9,10,11,12,13] and FCOS [14] and RetinaNet [15]. The YOLOv1 [10] model is proposed to simplify the object detection process to an end-to-end regression problem and directly predict the bounding box based on anchor box and complete the classification and location of the object. In 2016, an SSD [16] detection algorithm was proposed, which uses VGG16 [17] as the network structure and multi-scale feature map to detect objects, promoting the detection of small objects. In 2017, YOLOv2 [11] proposed to replace the original backbone network GoogLeNet [18] in YOLOv1 with DarkNet-19. In 2018, [12] proposed YOLOv3, which is widely used in the industrial field, to replace the network with DarkNet-53 [19], and added FPN (Feature Pyramid Networks) [20] to achieve multiscale training. In 2020, YOLOv4 [9] was proposed on the basis of YOLO series, CSPDarknet-53 was introduced as the feature extraction network, and SPP (spatial pyramid pooling) [21] was used in the feature extraction network to improve the effect of feature extraction, Mish [22] activation function was also used in the network, CIOU [23] was used to calculate bounding box loss, Mosaic was used for data augmentation, and PANet [24] network was used to integrate features of different scales, so as to further fusion the feature information of different layers. In 2020, Ultralytics [13] opened the code of YOLOv5.

Target tracking is to achieve the goal of tracking by analyzing the trajectory characteristics of the target [25]. Bewley et al. proposed the SORT algorithm [26] in 2016, combining the two-stage detection algorithm with Faster R-CNN as the detector. In 2017, Bewley et al. improved the SORT algorithm and proposed the DeepSORT [27] tracking algorithm. Based on the original SORT [26] algorithm, they proposed to combine the motion and appearance information of the target to track. In 2022, the StrongSORT [28] tracking algorithm was proposed, and two lightweight plug-and-play algorithms, AFLink model and Gaussian Smoothed Interpolation, were proposed. The feature library and cascade matching in DeepSORT algorithm were replaced by [29] feature extraction and vanilla global linear matching strategy.

In the helmet-wearing tracking experiment based on YOLOv5 and DeepSORT proposed by [30], it takes not wearing helmets as the tracking task. However, when the scene is complex, it is difficult to detect the target without a helmet and also easy to lose the target trajectory, so here we consider the person wearing a helmet as the tracking task. In [31], a tracking framework using YOLOv3 combined with DeepSORT was proposed to improve the YOLOv3 network architecture, design a deeper network, and introduce a new tracker to reduce ID switching. However, the network is complex in computing and is not suitable for helmet detection scenarios. The safety helmet detection scene is complex, the construction scene is frequently switched, the movement track is complex and fuzzy, and the movement scale changes frequently. In [32], YOLOv3-Tiny and YOLOv4 combined with DeepSORT, respectively, are proposed to complete the vehicle pedestrian tracking task. However, the performance of the YOLOv3-Tiny detector is not as good as YOLOv5, and the YOLOv4 model is too large. In addition, the ID of the DeepSORT tracker is more easily lost when the scene is complex.

The difficulties in monitoring the wearing of helmets lie in the complex trajectory of the target, the presence of target occlusion and the change of target attitude. These factors will lead to incorrect monitoring and missing detection of the target, or ID loss and ID switching when tracking whether a helmet is being worn, which will affect the real-time monitoring results. Combined with the above problems, this paper takes the construction personnel wearing helmets as the monitoring object, and processes more than 8000 images as the training dataset. YOLOv5 is selected as the detector for training, and the StrongSORT [28] tracking algorithm with good performance is used to achieve target tracking, so as to achieve the monitoring requirement of wearing a safety helmet under the construction background.

This paper uses detection-based tracking, combined with better detector and tracker algorithms, to achieve the task of tracking and monitoring whether a helmet is being worn in real-time scenarios.

## 2. YOLOv5

In the detection-based tracking task, the most important step is to select an appropriate detector, and the result trained by the detector model directly affects the effect of target trajectory tracking. The detection speed and detection precision of the object detector also directly affect the real-time tracking of the target trajectory. In YOLOv5, there are four network models, named YOLOv5s, YOLOv5m, YOLOv5l and YOLOv5x, respectively. In this paper, the YOLOv5s model is selected as the detector model in the tracking task.

### 2.1. YOLOv5s

The YOLOv5s network structure mainly includes the Input and Backbone network and Neck network and Prediction header, as shown in Figure 1. In the input part, the input size is set to 640 × 640 × 3, and the input image will use the Mosaic method for data enhancement. The main idea is to randomly crop any four images in the dataset, and then splice them as the training image, so as to achieve the effect of enriching the dataset. An adaptive anchor box calculation is introduced in the input section, the optimal initial anchor box size is selected for different datasets, and the image is processed with adaptive image scaling, reducing calculation time and improving the performance of detection. In the backbone section, the BottleneckCSP network structure is used to separate features by separating channels, and stitching when predicting output, which reduces the repeated calculation of feature information in the calculation process and enhances the ability of CNN to learn more feature information. In the Neck section, the structure of FPN and PANet is used. In the head section, convolution of three feature layers of different scales of the Neck layer makes the final prediction output.

### 2.2. Bounding Box Regression Loss

The traditional IOU (intersection over union) [33] calculates the overlap rate of the predicted box and ground truth box, that is, the ratio of their intersection and union. In this kind of bounding box regression algorithm, when the prediction box and the ground truth box do not have an intersection, the IOU loss remains at 0, degenerates to a constant, the regression loss of the predicted bounding box cannot be measured, and the convergence rate of the IOU loss is very slow. Therefore, Rezatofighi, H. et al. [34] proposed generalized intersection over union, GIOU, to solve the problem of traditional IOU loss, but when the prediction box and the ground truth box coincide, GIOU degenerated into traditional IOU loss, and the above problems will also occur. Therefore, on the basis of GIOU, combining the proportion of the non-cross area and the proportion of the distance between the center point of the two boxes, Zheng, Z. et al. [23] proposed DIOU (distance-IOU loss), which solves the situation when the two boxes of the prediction box and the ground truth box are contained horizontally and vertically with each other. When the prediction box coincides with the center point of the ground truth box, DIOU degenerates into the traditional IOU form. Therefore, complete-IOU (CIOU) loss [23] is proposed, combined with the aspect ratio penalty to deal with the problem of center point coincidence. As a result, CIOU takes into account both the loss of the cross area and the loss of the center point offset, combined with the loss of the proportion of width and height. Therefore, in the YOLOv5s detection algorithm, CIOU is used in the region proposals regression, and the calculation loss function is shown in Equation (1).
(1)$$L_{CIOU}=1-IOU+\frac{\rho^{2}\;\left(b,b^{gt}\right)}{c^{2}}+\alpha v$$

#### Focal-EIOU

CIOU considers various problems in the regression process of the prediction box and accelerates the convergence of the prediction box; however, there will be some problems that the width and height of the prediction box cannot be converged in a certain proportion at the same time. The EIOU [35] loss considered these problems and its calculation form is shown in Equation (2).
(2)$$L_{EIOU}=L_{IOU}+L_{dis}+L_{asp}=1-IOU+\frac{\rho^{2\;}\left(b,b^{gt}\right)}{{\left(w^{c}\right)}^{2}+{\left(h^{c}\right)}^{2}}+\frac{\rho^{2\;}\left(w,w^{gt}\right)}{{\left(w^{c}\right)}^{2}}+\frac{\rho^{2\;}\left(h,h^{gt}\right)}{{\left(h^{c}\right)}^{2}}$$

In the prediction of bounding box regression, the training samples play an important role in the convergence process. Therefore, under the premise of EIOU loss, this paper uses Focal-EIOU [35] to replace the prediction box regression loss CIOU in the original YOLOv5, and adds the inhibitory factor γ to promote the training samples with more information to play more roles in the regression process; its calculation form is shown in Equation (3).
(3)$$L_{Focal-EIOU}=IOU^{\gamma }L_{EIOU}$$

After replacing the CIOU loss of the bounding box regression with the Focal-EIOU loss, training the YOLOv5s model, the mAP@0.5 of all classes on the validation dataset is improved, which increased from 95.1% to 95.4%, mAP@0.5:0.95 increased from 62.1% to 62.9%, and the detection precision of wearing helmet increased to 96.5%, which were increased by 0.3%, 0.8% and 0.8%, respectively.

## 3. StrongSORT

The SORT [26] algorithm for multi-object tracking combines past video frames and current video frames, and solves the problem of prediction of object motion trajectory and video frame data correlation through Kalman Filter [36] and Hungarian algorithm, respectively, so as to achieve correlation detection of video frames. However, when the object is occluded, the next video frame predicted by Kalman Filter and the result detected by the detector will fail to match, and the trajectory tracking of that object will end, leading to ID switching of a large number of targets. DeepSORT [27] adds a pre-trained CNN network to the SORT to save the appearance features of each trajectory of the last 100 frames, solving the problem of ID switching caused by the object due to occlusion. At the same time, DeepSORT introduces cascade matching and new trajectory confirmation to improve the optimal matching of the predicted trajectory with the object in the current frame. Du, Y. et al. [28] proposed StrongSORT; two plug-and-play lightweight algorithms are introduced: AFLink and GSI. Among them, the AFLink model associates the short trajectory as a complete trajectory, which is a fully connected model without appearance information, GSI improves the absence in detection by simulating nonlinear motion, achieves more accurate positioning based on Gaussian regression, and does not ignore the motion information of the detected object during the regression process.

### 3.1. Kalman Filter

In [27], the motion information $(u,v,\gamma ,h,\dot{x},\dot{y},\dot{\gamma },\dot{h}$) of the object is described by an eight-dimensional space, which are the center coordinates (u,v) of the bounding box, respectively. Taking the aspect ratio γ, height h, and the relative speed of each variable in the object image coordinates, Kalman Filter is used to predict and update the object trajectory in the next frame, and the state $X_{i}$ at time t−1 is used to predict and transfer, and according to Equation (4), we obtain the status information X at time t, where F is the state transition matrix. The error of these two states is represented by the covariance matrix $P_{k}$, then the state error at the next moment t is described as $P_{t}$, as shown in Equation (5), to complete the prediction of the track state information of the next frame.
(4)$$X=FX_{t-1}$$
(5)$$P_{t}=FP_{t-1}F^{T}+Q$$

After completing the trajectory prediction of the next frame, the Hungarian algorithm is used to match the predicted trajectory with the object detected in the current frame, and a Kalman Filter is used to update the trajectory of the successful matching.

### 3.2. Cascade Matching

In [26], a Kalman Filter updates the predicted next frame trajectory result and matches the detected object. In [27], the trajectory predicted by Kalman is divided into a confirmed state and an unconfirmed state, and the newly predicted trajectory is initialized to an unconfirmed state; the Hungarian algorithm matches the detected object to a certain number of times before it is converted into a confirmation state, and the trajectory of the confirmation state will be matched with the detected box detected by the detector. In cascade matching, first, the set of deterministic trajectories predicted by the Kalman Filter is denoted as T, and the set of detected boxes is denoted as D, and the cost matrix C of the two is calculated by Equation (6); the Mahalanobis Distance is used to describe the trajectory predicted by Kalman Filter and the motion information of the current detection box,
(6)$$c_{\left(i,j\right)}=\lambda d^{\left(1\right)}\left(i,j\right)+\left(1-\lambda \right)d^{\left(2\right)}\left(i,j\right)$$
which is d(i,j). As shown in Equation (7), $d_{j}$ represents the j–th detection box.
(7)$$d^{\left(1\right)}\left(i,j\right)={\left(d_{j}-y_{i}\right)}^{T}S_{i}^{-1}\left(d_{j}-y_{i}\right)$$

$\left(y_{i},S_{i}\right)$ represents the projection of the i–th trajectory to the detection space. Second, the matches that do not conform to the Mahalanobis Distance are removed according to the set threshold (as in Equation (8)).
(8)$$b_{i,j}=\prod_{m=1}^{2}b_{i,j}^{\left(m\right)}$$

Finally, according to the update status of the prediction box, the newer prediction box will be matched with the Hungarian algorithm first.

### 3.3. AFLink

Over-reliance on the feature information of the object is easily affected by noise. The pursuit of high performance and detection speed by correlating global trajectory information will result in complex computations and a large number of hyperparameters. AFLink directly predicts the association of two trajectories through time information. In the AFLink model, trajectories $T_{i}$ and $T_{j}$ are used, as shown in Figure 2 for the AFLink model structure, where $T_{*}={\left\{f_{k},x_{k},y_{k}\right\}}_{k=1}^{N}$ consists of the position information of the last 30 frames and the frame $f_{k}$. $T_{i}$ and $T_{j}$ will be input into the time module and the fusion module [28]. The time module is used to extract frame feature information, and then the fusion module is used to perform feature fusion on the extracted frames of different dimensions, and then the classifier predicts the correlation between the two frames. In this process, the two trajectories $T_{i}$ and $T_{j}$ do not interfere with each other in the processing of the time extraction module and the fusion module.

### 3.4. Appearance Information

For appearance feature information, in DeepSORT, a CNN network pretrained on the pedestrian re-identification dataset is used, the CNN is used to extract pedestrian features, and the pedestrian features are saved, and the CNN feature extractor is used for pedestrian tracking tasks. When using [27] to track objects, it will save the features of the 100 most recent frames of each track in a feature library. When a video frame has an undetected object, it calculates the feature library $R_{i}$ of the i–th track and the j–th detection. The minimum cosine distance of the feature $f_{j}$ of the object is shown in Equation (9).
(9)$$d\left(i,j\right)=\min\{1-f_{j}^{T}f_{k}^{\left(i\right)}\;|f_{k}^{\left(i\right)}\in R_{i}\}$$

In StrongSORT, the CNN that extracts feature information is replaced by a feature extractor BoT network, which can extract more feature information about the detection object in the video frame. At the same time, the feature library extracted and saved by CNN is changed to a feature update strategy, that is, the appearance state $e_{i}^{t}$ of the i–th track in the t–th frame is updated in the form of an exponential moving average, as Equation (10) shows.
(10)$$e_{i}^{t}=\alpha e_{i}^{t-1}+\left(1-\alpha \right)f_{i}^{t}$$
$f_{i}^{t}$ is the appearance embedding information of the detection matched by the current trajectory.

## 4. Experiments and Analysis

### 4.1. Construction of Dataset

The dataset used for experimental detection and tracking was selected from the Safety Helmet Wearing Dataset for the experiment; the dataset includes more than 8000 images, includes workers wearing and not wearing a helmet in various construction scenarios, and images of complex task-overlapping scenes, dark scenes, character occlusion and other scenarios. At the same time, some negative samples without helmet were added to the dataset to increase the difficulty of detection. The dataset sample is consistent with the real-time monitoring of the actual construction scene, so, in this paper, the Safety Helmet Dataset is selected for this detection and tracking task, and the dataset is processed into three categories, namely the target helmet to be detected and samples that not wearing a helmet and the head as a negative sample interference. Figure 3a–d contain positive and negative samples, night samples, character occlusion, and negative samples, respectively.

We recorded a short video at our school construction site as a real-time detect video to test our model.

### 4.2. Experimental Environment

The specific training environment of our experiment is as shown in Table 1.

In this experiment, the mean of Average Precision, mAP and Frame Per Second, FPS are used as the evaluation criteria for the YOLOV5s model. The speed at which video frames are processed per second and the frequency at which the target ID is switched are used as metrics for the StrongSORT tracking model.

### 4.3. Experimental Results and Analysis

#### 4.3.1. The Results of YOLOv5s

In the experiment, the epoch is set to 300 and the batch size is 8. The results of training on the helmet dataset are shown in Figure 4a,b. As shown in Figure 4a, the training precision of wearing a helmet can reach 95.7%, the precision of not wearing a helmet can reach 94.4%, and the mAP@0.5 of all classes on the validation dataset can reach 95.1%. Figure 4b shows that there are no false detections and missed detections.

The results of the loss in the training process can be seen, as shown in Figure 5; after the class loss on the validation dataset is trained to 200 epochs, the loss begins to converge and remains at 0.0015 until the 300 epochs of iterations are completed.

As can be seen in Figure 6, the classification loss during the training process can be reduced to an effect of 0.00018.

At the same time, we use YOLOv5s’s best weight result to detect real-time recorded construction site videos. Figure 7 shows the results of missed detection and full detection in different video frames. In Figure 7a, a small number of occluded objects cannot be detected. In Figure 7b, it can be seen that all objects are detected, and the detection of video frames can reach an average inference speed of 16.2 ms. The average detection processing speed of each frame is 0.015 s.

#### 4.3.2. Tracking Results of StrongSORT

After using the YOLOv5s model to train the dataset, the target person wearing a helmet can be basically detected. The trained detector is combined with the StrongSORT to achieve the effect of real-time target tracking. As shown in Figure 8, the upper left corner is the unique identification (ID) number of the target person, Figure 8a–c show the tracking situation of the same target person in different video frames by the StrongSORT, and there is no target ID switching, and there is no false detection in real-time monitoring and tracking. Taking the target person whose ID is 17 as an example, in different video frames, although the target is occluded for a long time, the ID of the target person has never been switched, which achieves a good tracking result. In Figure 8c, the target without a helmet is not marked as a positive sample, and no false detection occurs.

Using StrongSORT to achieve tracking, it achieves a processing speed of 26.5 ms per frame. At the same time, the average detection speed of YOLOv5s is 0.017 s, and the average speed of StrongSORT to update the video frame track is 0.05 s.

#### 4.3.3. Detector Comparison Experiment

In this paper, the two-stage detection models Faster R-CNN + FPN, Cascade Masked R-CNN + FPN, and the one-stage detection model YOLOv3 + SPP are compared with the improved YOLOv5 detector model, and each model uses pretrained weights during training. Under the same conditions, the above-mentioned Safety Helmet Dataset is used for training, and the weight size, mAP@0.5, FPS, and the saved weights model size is used as the metrics for evaluating the detector. The experimental results are shown in Table 2, and it can be seen that the detection model proposed in this paper has reached 95.4% of the mAP@0.5 of all categories on the validation dataset, and the precision of wearing helmets can reach 96.5%, as illustrated in Figure 9. The inference speed FPS has reached 100 images per second, and the model weight is only 14.4 MB, which is better than other detection models. The original YOLOv5s network weight size was 14.5 MB, and the weight model size was not much different after changing the prediction box regression loss.

#### 4.3.4. Tracker Comparison Experiment

In addition, the use of DeepSORT [27] and StrongSORT [28] algorithms are selected in this paper. The above algorithms are combined with the YOLOv5s + Focal-EIOU detector for comparative experiments. We use the character occlusion ID switching and FPS as evaluation indicators. As shown in Table 3, the StrongSORT processes tracking and detection is faster than DeepSORT, reaching 37 frames per second.

In the experimental results of YOLOv5s combined with DeepSORT, the phenomenon of frequent switching of the target ID after the target occlusion occurs, as shown in Figure 10, where Figure 10a,c are the same target ID at different frame moments, the ID is switched from 858 to 939 due to the target person being obscured, and in Figure 10b the ID is lost because the target is obscured. In YOLOv5s combined with StrongSORT, for the same target person, there is no ID switching or ID loss when the targets have been obscured, which has always been 223, as shown in Figure 10d,e.

## 5. Conclusions

In this paper, the object detection model YOLOv5s is combined with the tracking algorithm StrongSORT [28] to realize the tracking of helmet wearing. According to the comparative experiments, the YOLOv5s model is the most suitable choice in terms of detection speed and detection precision. In addition, in the tracking comparison experiment, StrongSORT [28] has a faster processing speed than DeepSORT [27], and the target ID will not be lost or switched due to problems such as long-term occlusion and large changes in motion scale, amongst the other reasons. At the same time, the speed of processing detection and tracking has also achieved good results. In the next work, we will explore how to apply this work to embedded terminal applications.

## Figures and Tables

**Figure 1 sensors-23-01682-f001:**
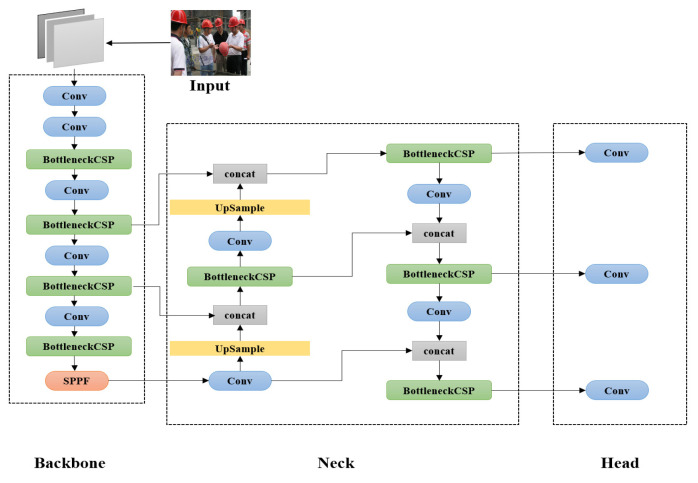
The network model of YOLOV5s.

**Figure 2 sensors-23-01682-f002:**
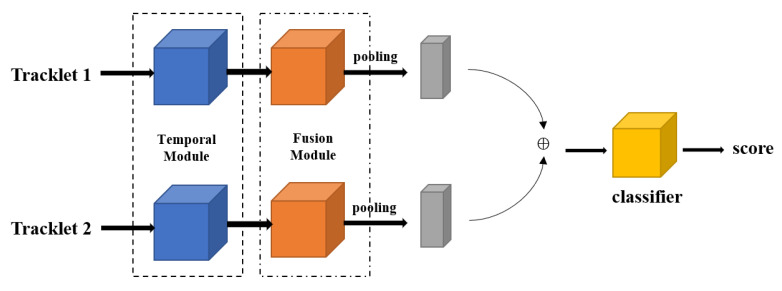
The network model of AFLink.

**Figure 3 sensors-23-01682-f003:**
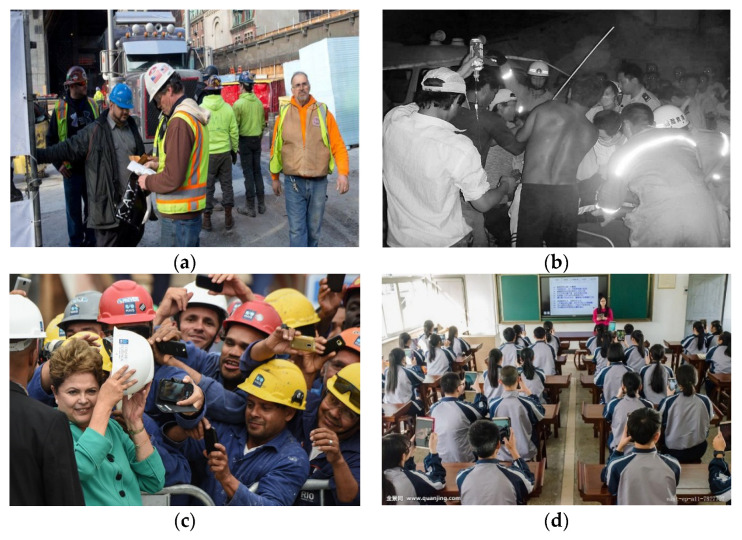
Four different test scenarios with helmets. (**a**) including positive and negative samples; (**b**) dark samples; (**c**) object occlusion; (**d**) negative samples.

**Figure 4 sensors-23-01682-f004:**
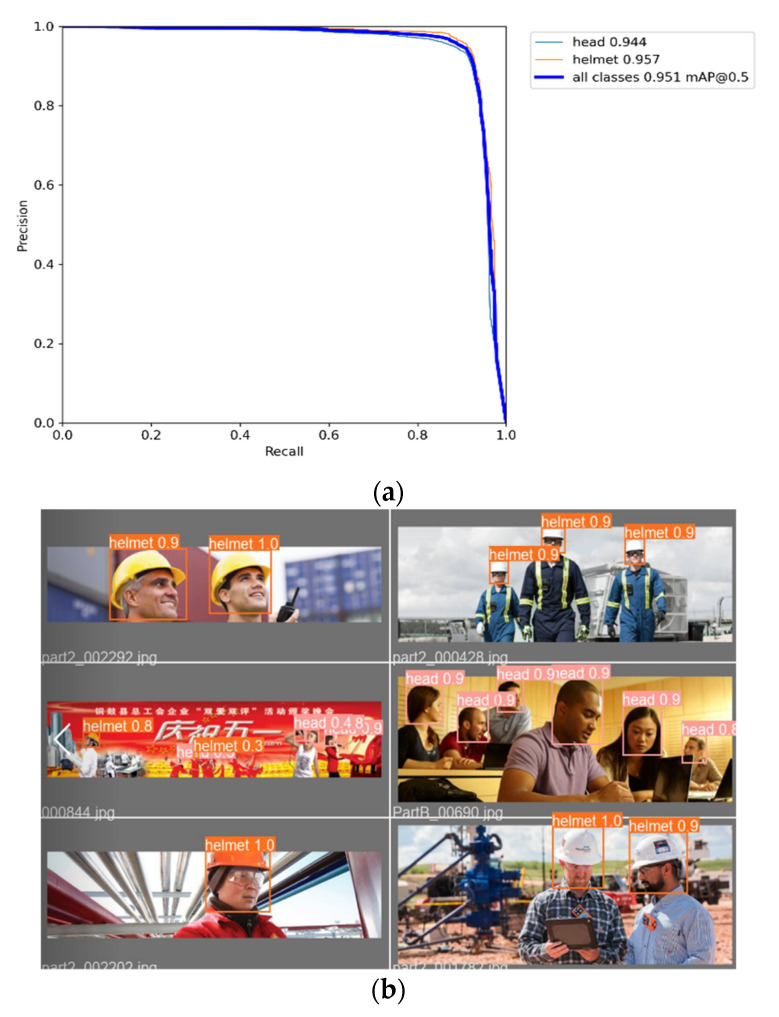
Training results of Safety Helmet Dataset on YOLOv5s. (**a**) Precision and Recall; (**b**) Training results of some samples.

**Figure 5 sensors-23-01682-f005:**
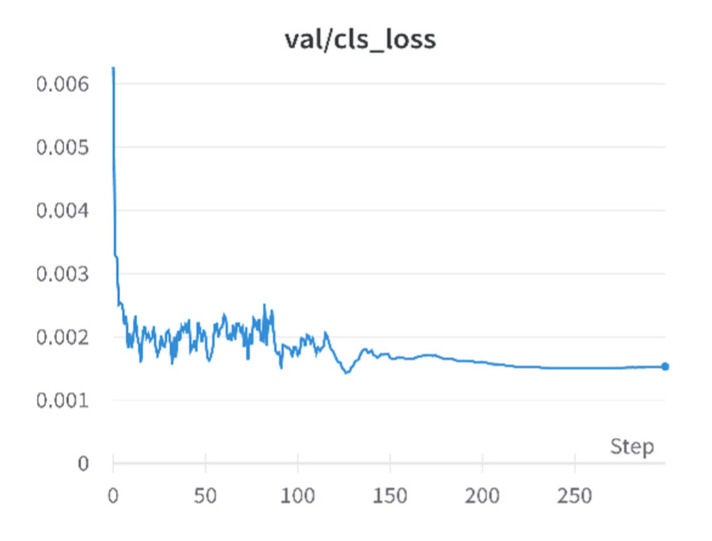
Validation dataset classification loss.

**Figure 6 sensors-23-01682-f006:**
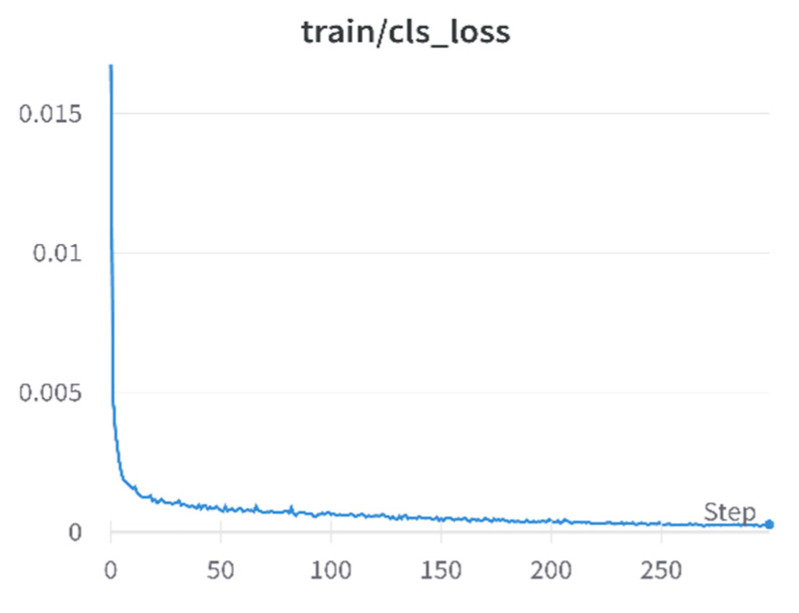
Training set classification loss.

**Figure 7 sensors-23-01682-f007:**
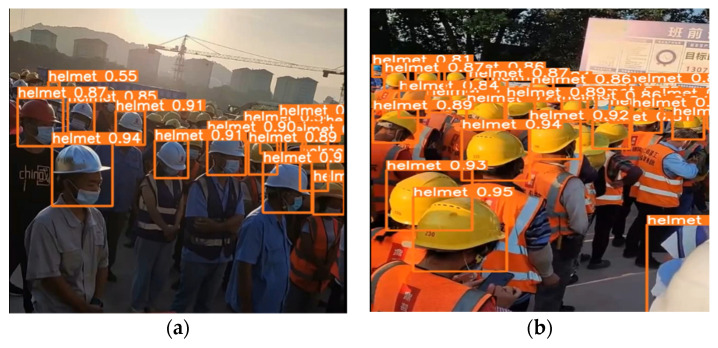
YOLOv5s video detection results. (**a**) missing detection; (**b**) full detection.

**Figure 8 sensors-23-01682-f008:**
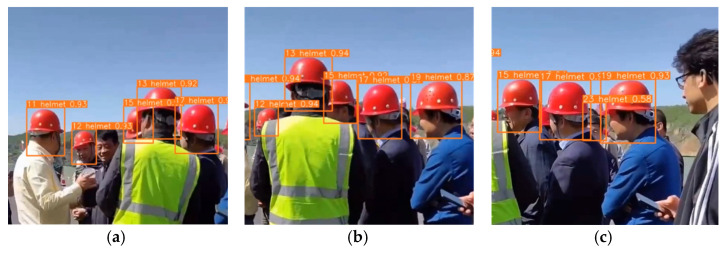
Tracking results of the same object person. (**a**) 10th second; (**b**) 15th second; (**c**) 25th second.

**Figure 9 sensors-23-01682-f009:**
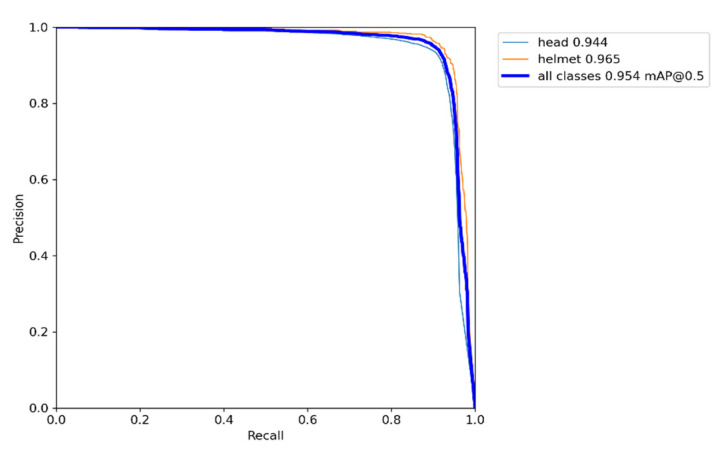
Precision and recall with focal-EIOU.

**Figure 10 sensors-23-01682-f010:**
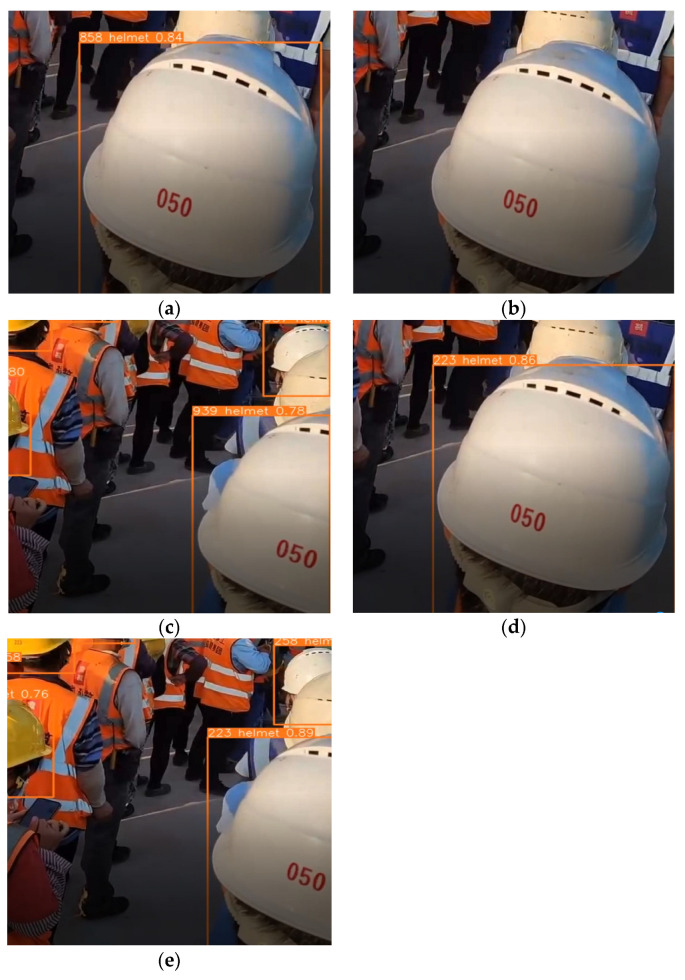
ID Status of same person in different frames in the video. (**a**) Frame 612; (**b**) Frame 689; (**c**) Frame 710; (**d**) Frame 670; (**e**) Frame 710.

**Table 1 sensors-23-01682-t001:** Training Environment.

Items	Version
Graphics	NVIDIA GeForce RTX 3070
Frame	Pytorch
System CUDA	Ubuntu 18.04 11.04

**Table 2 sensors-23-01682-t002:** Comparison experiment of detector.

Detector	mAP@0.5/%	FPS	Weight (MB)
Faster R-CNN + FPN	85.1	24	330.4
Cascade Masked R-CNN + FPN	85.5	19	552.8
YOLOv3 + SPP	87.9	55	338.8
YOLOv5s	95.1	77	14.5
YOLOv5s + Focal-EIOU	95.4	100	14.4

**Table 3 sensors-23-01682-t003:** Comparison of FPS.

Model	FPS
YOLOv5 + DeepSORT	34
YOLOv5 + StrongSORT	37

## Data Availability

Please contact the corresponding author for available data support.

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
