# Peer review of "Helmet-Wearing Tracking Detection Based on StrongSORT"

_sensors, 2023, doi:10.3390/s23031682_

Round 1

Reviewer 1 Report

The authors propose a framework model based on YOLOv5, combined with multi-object tracking algorithms, to monitor and track whether construction workers wear safety helmets in real-time video. The work is very interesting as is solves a real problem that is realted to the security of the works in a construction site.

Why do you consider Figure 3 (c) in the object occlusion category? Most of the workers in Figure 3 (c) are not occluded. What basically differs this image from the others is the distance of the workers to the camera. Please make this clear in the text.

I would be nice to have more details regarding the datasets used. For instance, how data is annotated? is there a bounding box that comprises the entire person or just the face? is there any case of images with just the helmet in it, without the person? in such case, what happens? how the data is annotated?

The authors indeed showed good performance results with the proposed algorithm combination, but it is too dangerous to claim that the proposed work is suitable to run in embedded devices, since the algorithm was validated using a NVIDIA GeForce RTX 3070 with 8GB of RAM memory. What would be the performance of the algorithm if it runs in an embedded device, such as an NVIDIA Jetson Nano, for instance? It would be nice to have information about the expected FPS in this embedded platform (or similar) because this information would strengthen the work.

Some general comments and writing errors found are listes as follows.

"Object detection algorithms based" -> "Object detection based"

"in the construction" -> "in construction"

"by safety helmets""-> "by the lack of safety helmets"

"On the one hand," -> "On one hand,"

"an effective ways" -> "an effective way"

"Classical two-stage algorithms such as the proposed R-CNN series [4-6]." -> this sentence seems incomplete

"[4]" -> use authors names with citation

"[5,6]" -> use authors names with citation

"[11]" -> use authors names with citation

"Hungary" or "Hungarian" algorithm?

"DeepSORT respectively" -> "DeepSORT, respectively,"

"a appropriate" -> "an appropriate"

"models, name YOLOv5s," -> "models, named YOLOv5s,"

"YOLOv5x respectively." -> "YOLOv5x, respectively."

what do you mean by "the input image has been enhanced by Mosaic"

"the four images are spliced together by random scaling," -> which four images?

"[35]" -> use authors names with citation

"[27]" -> use authors names with citation

"the CIOU [27]" -> "CIOU [27]"

"and its calculation" -> ", its calculation"

", to promote" -> "to promote"

"0.8% respectively." -> "0.8%, respectively."

"[17]" -> use authors names with citation

"?information" -> "? information"

"and we get the range of state ?information at time ? is shown in formula 4" -> please rewrite

"set of detected box" -> "set of detected boxes"

",as the equation 7 shown," -> ". As shown in equation 7,"

"Finally, the prediction results of the recent Kalman Filter are matched according to the cost matrix ? calculated above for Hungarian precedence, in short, match the most recently predicted frames first." -> please rewrite

"Appearance And Motion" -> "Appearance and Motion"

"calculate the feature" -> "it calculates the feature"

"object, as shown" -> "object is shown"

"[17]" -> use authors names with citation

"[17]" -> use authors names with citation

"trajectory. And replace" -> "trajectory and replace"

"Experiments and analysis" -> "Experiments and Analysis"

"Construction of dataset" -> "Construction of Dataset"

"is used to manually record construction scenes around the 331 school," -> which school?

"Experimental environment" -> "Experimental Environment"

"8G," -> "8GB,"

"This experiment uses GPU for accelerated computing processing, the specification is NVIDIA GeForce RXT 3070 8G, the model construction adopts the pytorch framework, combined with CUDA for parallel computing, and the specific training environment is as shown in Table 1." -> every information in this sentence is already detailed in Table 1

"Experimental results and Analysis" -> "Experimental Results and Analysis"

"(b), as shown" -> "(b). As shown"

"of not wear helmet" -> "of not wearing helmets"

"Figure (b)" -> "Figure 4 (b)"

"Figure 7 (a), (b)" -> "Figure 7"

"of occluded object" -> "of occluded objects"

"that all object are" -> "that all objects are"

"Yolov5s" -> "YOLOv5s"

"dataset and the target" -> "dataset, the target"

"the [17] tracking algorithm" -> please rewrite

"[17] algorithm," -> please rewrite

"[17]" -> use authors names with citation

"is 0.05s。" -> "is 0.05s."

"detector model in this paper," -> "detector model,"

"100 image per second," -> "100 images per second,"

"Tracker comparison experiment" -> "Tracker Comparison Experiment"

"[16] and [17] tracking algorithm" -> please rewrite

"In addition, the use of [16] and [17] tracking algorithm are selected in this paper combined with the YOLOv5s+Focal-EIOU detector model for comparative experiments, the character occlusion ID switching and FPS evaluation indicators, FPS comparison results as shown in table 3, it’s obvious that the model of YOLOv5s combined with [17] processing speed of video frames is faster, reaching 37 frames per second." -> please rewrite using more than one sentence

"it’s obvious that the model of YOLOv5s combined with [17] processing speed of video frames is faster" -> why is this obvious?

"[16]" -> use authors names with citation

"where (a) (c) is the same" -> "where (a) and (c) are the same"

"and (b) the ID" -> "and in (b) the ID"

"[17]" -> use authors names with citation

"the same target" -> "for the same target"

"in (d) (e) in" -> "in (d) and (e) in"

"in different frame" -> "in different frames"

"the trained model is small and suitable for use in embedded devices" -> how small should a model be to be suitable for embedded devices?

"[17] has a faster processing speed than [16]," -> use authors names with citation

"At the same time, the speed of processing detection and tracking has also achieved good results, which meets the requirements of practical application construction scenarios and is suitable for deployment in real-time monitoring systems." -> please make clear what are the requirements of practical application construction scenarios

"RXT" -> "RTX"
